# Magnetic Resonance Detects Structural Heart Disease in Patients with Frequent Ventricular Ectopy and Normal Echocardiographic Findings

**DOI:** 10.3390/diagnostics11081505

**Published:** 2021-08-20

**Authors:** Raffaele Scorza, Anders Jansson, Peder Sörensson, Mårten Rosenqvist, Viveka Frykman

**Affiliations:** 1Cardiovascular Unit, Department of Clinical Sciences, Danderyd University Hospital, Karolinska Institutet, 18288 Stockholm, Sweden; anders.r.jansson@sll.se (A.J.); marten.rosenqvist@sll.se (M.R.); viveka.frykman-kull@sll.se (V.F.); 2Department of Medicine, Solna, Karolinska University Hospital, Karolinska Institutet, 17177 Stockholm, Sweden; peder.sorensson@sll.se

**Keywords:** premature ventricular contractions, ventricular arrhythmia, cardiac magnetic resonance, cardiovascular imaging

## Abstract

The prognosis of patients with ventricular ectopy and a normal heart, as evaluated by echocardiography, is virtually unknown. Cardiac magnetic resonance (CMR) can detect focal ventricular anomalies that could act as a possible site of origin for premature ventricular contractions (PVCs). The aim of this study was to investigate the presence of cardiac anomalies in patients with normal findings at echocardiogram. Methods: Fifty-one consecutive patients (23 women, 28 men, mean age 59 years) with very high PVC burden (>10,000 PVC/day) and normal findings at standard echocardiography and exercise test were examined with CMR. The outcome was pathologic findings, defined as impaired ejection fraction, regional wall motion abnormalities, abnormal ventricular volume, myocardial edema and fibrosis. Results: Sixteen out of 51 patients (32%) had structural ventricular abnormalities at CMR. In five patients CMR showed impairment of the left ventricular and/or right ventricular systolic function, and six patients had a dilated left and/or right ventricle. Regional wall motion abnormalities were seen in six patients and fibrosis in four. No patient had CMR signs of edema or met CMR criteria for arrhythmogenic right ventricular cardiomyopathy. Five patients had extra-ventricular findings (enlarged atria in three cases, enlarged thoracic aorta in one case and pericardial effusion in one case). Conclusions: In this study 16 out of 51 patients with a high PVC burden and normal findings at echocardiography showed signs of pathology in the ventricles with CMR. These findings indicate that CMR should be considered in evaluating patients with a high PVC burden and a normal standard investigation.

## 1. Introduction

Premature ventricular complexes (PVCs) are common clinical findings, both in patients with and without structural heart disease, and the reported prevalence, although it varies among different studies, is high [1,2,3,4,5,6] Whereas PVCs are known to be harbingers of poor prognosis in people with previous or current cardiac pathology, there is no agreement about their prognostic impact on subjects without known heart disease [7,8]. Furthermore, questions have been raised about whether individuals with a high PVC burden and normal findings at echocardiography are really free of structural heart disease or if actually more advanced imaging methods are required to identify subtle disease that PVCs could cause or from which they may ensue [9].

While the general recommendations in the diagnostic pathway for PVC patients include echocardiography [10], in 1997 it was shown that cardiovascular magnetic resonance (CMR) could add significant information [11]. Current ESC guidelines for the management of ventricular arrhythmias assign a IIa recommendation for CMR in evaluation of PVC patients [12], referring to the previous joint recommendation from ESC/ACC/AHA [13]. AHA published new guidelines in 2017, again assigning a IIa recommendation that patients with frequent ventricular ectopy should be evaluated with CMR to detect underlying heart disease [14]. However, the authors did not specify what this recommendation was based on. Furthermore, the published studies have diverse designs and inclusion criteria. Most of them include only patients with a specific PVC morphology, are based on a selected population, or in other ways do not include consecutive patients [11,15,16,17,18,19,20,21,22,23,24].

The aim of this study was to assess the occurrence of structural pathology at CMR in a cohort of consecutive patients with a high burden of PVCs and normal findings at echocardiography and exercise test, regardless of the PVC morphology.

## 2. Methods

We prospectively included 51 consecutive patients between 2016 and 2018 with at least 10,000 PVCs per day according to Holter recording. All the patients were evaluated at our Arrhythmia Outpatient Clinic by an experienced cardiologist and underwent an exercise test and echocardiography with a normal result. No patient had a history of previous structural heart disease or sustained ventricular tachycardia. In a majority (42 of 51) of patients we also had access to PVC on 12-lead ECG. The only exclusion criteria were contraindications to CMR or inability to perform the exam (e.g., claustrophobia).

The Holter data were analyzed manually by an experienced physician to ensure accuracy. The recording time was at least 24 h, and the first 24 h were considered for inclusion.

A normal exercise test was defined as absence of exercise-induced depression of the ST segment on ECG and absence of exercise-induced arrhythmia (if ventricular arrhythmia was present before exercise and diminished or stayed constant during exercise, the result was not considered pathologic).

A normal echocardiogram was defined as left ventricular ejection fraction (LVEF) equal to or higher than 55%, visually normal right ventricular ejection fraction (RVEF), absence of moderate to severe valve dysfunction, absence of local dyskinesia, normal ventricular dimension, and normal wall thickness. Echocardiographic exams were reviewed by two additional independent and blinded examiners. In case of conflicting evaluations of the echocardiographic findings, the exam was considered normal if two of three examiners had assessed the exam as such, and pathologic if two of the three examiners had found signs of pathology. As a result of this process, we excluded one of the 52 originally included patients.

CMR examinations were performed using a 1.5 T Signa HDxt scanner (GE, Milwaukee, WI, USA) with a cardiac phased array 32 channel coil out of which 16 channels were used. All patients were in sinus rhythm during the investigation. For the assessment of regional wall motion abnormalities (RWMA) and left ventricular (LV) and right ventricular (RV) volumes, cine images were used with a steady-state free precession (FIESTA) cine sequence in long axis views and short-axis stack (from the right ventricular outflow tract to the apex, 8-mm slice thickness, 2 mm gap). In patients with PVCs of unknown QRS morphology or morphology suggestive of RV origin a transaxial cine stack (from diaphragm to the pulmonary bifurcation, 8-mm slice thickness, no gap), sagittal RVOT and RV in-/outflow views were added. The following acquisition parameters were typically applied: 30 phases, 24 views per segment adjusted for heart rate, NEX 1, FOV 35 cm, a matrix of 256 × 224, a 40–45 °C flip angle, TR/TE approximately equal to 3.5/1.5, and a bandwidth of 125 kHz.

For evaluation of edema, black blood T2w images (STIR) were performed in long axes views and short axis stack (12-mm slice thickness, 8 mm gap).

Late gadolinium enhancement (LGE) was performed using a gadolinium dose of 0.2 mmol/kg bodyweight, but decreased to 0.1 mmol/kg in patients with GFR 30–60 mL/min. For LGE a standard 2D–IR GRE sequence was performed in long and short axis views using the same slice thickness and gap as for SSFP cines.

We used dedicated software (Segment CMR, Medviso, Lund, Sweden) for post-processing, and functional parameters were obtained from the short-axis images. For both LV and RV, end-diastolic and end-systolic volume indexes and ejection fraction were reported and compared to the respective reference values for class age and sex [25]. Segmental LV and RV WM abnormalities were investigated from all available cine images and reported as hypokinetic, akinetic or aneurysmal. If edema or LGE was present the segmental, and for LGE transmural (subendocardial, mid wall, transmural), extent was reported. Two expert investigators (one with EACVI level III expertise) who were blinded from one another’s opinion evaluated each examination; their consensus was sought in case of inconsistency.

Pathology at CMR was defined as one or more of the following: abnormal LV or RV volume, abnormal wall thickness, regional dyskinesia, myocardial edema, fibrosis and ejection fraction (EF) lower than 55%.

## 3. Statistics

No power calculation was done before inclusion, as this is regarded as a descriptive pilot study. Calculation of mean values was done in Excel (Microsoft, Redmond, WA, USA). Categorical variables were compared using Chi-squared with Yates’ correction. A two-sided *p*-value of ≤0.05 was considered statistically significant. Continuous data were presented as mean ± standard deviation or median ± interquartile range (IQR) when appropriate. Nominal data are presented as number of cases (percent).

## 4. Results

A total of 51 patients, 23 females and 28 males, was included. The median age was 62 years (IQR = 44–73). Patient characteristics are summarized in Table 1. Despite the high PVC burden, no patient had a history of symptomatic ventricular tachycardia. During sinus rhythm, right bundle branch block was present in two patients, three had a left hemiblock, while one patient had both right bundle branch block and left hemiblock. The remaining patients had a normal QRS morphology.

Holter recording was available for all patients, while PVCs recorded on 12-lead ECG were available in 42 patients. We determined the PVC site of origin by assessing the direction of depolarization on the sagittal and the frontal plan, using a comprehensive paper from 2015 as guidance [26]. The morphologies from 42 patients are summarized in Figure 1. The number of PVCs during 24 h of recording ranged between 10,000 and 40,000 (mean = 18,800 ± 6865).

### 4.1. CMR—Findings

All patients completed the CMR protocol without complications. Sixteen patients (31%) had pathologic findings at CMR, with three patients (6%) having more than one pathological finding. The most common findings were RWMA, dilated left ventricle and fibrosis, while no patient had edema. The results are summarized in Figure 2 and the physiological parameters measured by CMR are in Table 2.

All cases with RMWA and dyskinetic areas were found in the right ventricle, while fibrosis was only seen in the left ventricle. In the two cases of fibrosis in which we had access to 12-lead ECG there was a correlation between topographic localization of the finding and the PVC morphology (Figure 3).

We ran a logistic regression with the number of PVCs as predictor variable and any CMR finding as the outcome variable, showing no statistically significant correlation (*p* = 0.44).

### 4.2. PVC Morphology

Of the three patients with solitary impaired left ventricular function, one had RVOT PVCs, one had multifocal PVCs, while the third patient had no 12-lead ECG recording of the PVCs. One patient showed biventricular dysfunction, and another one had solitary right ventricular dysfunction, both of these patients had RVOT PVCs.

Out of six patients with left ventricular dilation (one with biventricular dilation), we had PVC recorded on a 12-lead ECG in four, and found both a right ventricular and a left ventricular morphology in two cases.

The relationships between findings and PVC morphology are summarized in Table 3, Table 4 and Table 5. No PVC morphology was statistically associated to a higher prevalence of CMR findings, all calculated differences of proportions between groups yielding a *p*-value > 0.05.

CMR could also identify extraventricular pathology; three patients had enlarged atria (one of them had previously been diagnosed with atrial fibrillation), one had an enlarged thoracic aorta and one had pericardial exudate.

## 5. Discussion

In this study CMR showed signs of pathology in 16 of 51 patients (31%) with high PVC burden, despite normal findings at a standard investigation that included echocardiography and exercise test. The occurrence of CMR findings did not significantly vary with PVC morphology, but we did notice that focal or local findings such as fibrosis or wall motion abnormalities tended to be present in the same area as the site of origin of PVCs.

There is an increasing body of evidence recently summarized in comprehensive reviews [27,28,29] about the role of CMR in evaluation and risk-assessment of patients with ventricular arrhythmia. However, the studies behind this evidence were performed in different settings and populations, and a majority were carried out with restrictive inclusion criteria and in selected groups. In some studies the patient had not been evaluated before CMR with both the exercise test and echocardiogram [16,21,30,31], or only patients with specific PVC morphologies were included [32,33].

In our study we included consecutive patients regardless of PVC morphology, all with normal findings at a thorough standard examination. In our opinion, this design makes the study an important contribution to the evidence that has accrued since, when the first two studies were published in 1997 on patients with PVCs originating from RVOT [11,30]. Even later studies tended to focus on right ventricular arrhythmias, and ability in detecting focal pathology after normal findings at standard examinations [16,18,34]. However, other studies showed essentially normal CMR findings in patients with repetitive monomorphic right ventricular tachycardias [31,35].

A considerable proportion of our patients (45%) had unifocal PVCs originating from the right ventricle. Patients with RV–PVC have, both in clinical praxis and scientific papers, been the object of evaluation aimed at detecting diagnostic criteria for arrhythmogenic right ventricular cardiomyopathy (ARVC). However, even when diagnostic criteria for ARVC are not met, the CMR findings still may have a prognostic value. Aquaro et al. enrolled 440 patients with a burden of >1000 PVCs/day with a LBBB morphology. The participants were evaluated with CMR in search of ARVC criteria, other than PVC originating from the right ventricle. The study also had a clinical outcome consisting of a composite end point of cardiac death, resuscitated cardiac arrest and appropriate ICD shock. The patients with RV anomalies had worse clinical outcomes than patients without RV anomalies, despite a few patients with MRI anomalies fulfilling task criteria for ARVC diagnosis [15]. There was also a quantifiable effect of CMR findings on prognosis, as patients with more than one finding had worse outcome than those with only one finding.

PVC morphology was also important in the study from Nucifora [17], that compared groups with left ventricular and right ventricular PVCs. The LV group had significantly more anomalies at CMR than the RV group, and these anomalies were associated with worse clinical outcome (composite endpoint of sudden cardiac death, nonfatal ventricular fibrillation and appropriate defibrillator discharge).

The CMR findings in our study were subtle and no patients met diagnostic criteria for ARVC or other well-defined structural heart diseases. Four patients showed signs of fibrosis, but none was of ischemic origin (previous myocardial infarction) based on LGE pattern, localization and distribution. In a recently published study [36] the authors established an underlying diagnosis in about 30% of the unrolled patients, despite a normal echocardiogram, with an additional 20% of the patients showing more subtle signs of pathology.

Differently from other studies [17,18,21], we had a pre-defined cut-off value for numbers of PVCs at inclusion. Other studies [37] presented both a lower and an upper for PVC burden. However, we could present similar findings.

As previously mentioned, the opinion about the prognostic meaning of pathologic CMR findings is divided, with a majority of the few published articles claiming them to be a negative risk marker. It is currently uncertain whether PVCs themselves have a pathogenetic role or are merely markers of an active disease of the myocardium, even when no signs of structural pathology can be seen at imaging. It is reasonable to believe that both phenomena exist. The large multicenter study by Muser et al. [38] deserves mention, in which subjects with cardiac abnormalities at CMR (16% of the participants) had a worse prognosis compared to PVC patients without pathologic CMR findings.

Our findings revealed a possible relation between the site of fibrosis at CMR and PVC morphology, indicating that PVCs had their origin in the fibrotic area, which points toward them being an effect of a previous disease. However, in patients with impaired ventricular function or dilated ventricles, there was no correlation between the diseased ventricle and the origin of PVCs, possibly meaning that PVCs per se lead to unfavorable effects on ventricular functions. The mechanisms leading to these unfavorable effects are unknown, although several theories have been formulated including electromechanical dyssynchrony, extrasystolic potentiation, interpolation, R–R variability and myocardial remodeling by short-coupled PVCs [39]. Dyssynchrony has been highlighted in particular as a possible pathophysiologic mediator [40], an explanatory mechanism that PVC-induced cardiomyopathy could have in common with cardiomyopathy caused by ventricular pacing [41]. When this is performed from the right ventricular apex, it can affect cardiac function negatively over time.

Previous papers have shown that CMR can detect previously undiscovered ventricular anomalies in a significant share of the participants, ranging between 15–75% [19,23,32,38,42]. The findings in our study are consistent with studies with similar design, however, we believe our data can be considered highly generalizable, since they are based on consecutive patients from common clinical practice. Our findings indicate that CMR can have a roll in evaluating all patients with high PVC burden despite a normal echocardiography and regardless of PVC origin and morphology. There is evidence that pathological CMR findings implicate a negative clinical outcome, which makes studies including larger patient group and long follow-up necessary.

## 6. Limitations

This is a descriptive pilot study, and like most papers in the field, we did not include controls. We made this decision based on the recommendations from the International Society for Magnetic Resonance in Medicine [43], which urged caution in the use of gadolinium because it has been shown that it tends to deposit in brain tissue. Although this phenomenon is likely to be biologically harmless, we opted for a careful approach. In respect to the absence of controls, we also decided on a conservative approach to evaluating CMR findings; in the few situations when the two independent examiners evaluated the finding differently, we decided to interpret the finding as normal variation and not pathology.

As frequent PVCs can pose a challenge in evaluating CMR sequences, all our findings are based on frames with good quality. Frequent PVCs during CMR procedure may lead to false positive findings of myocardial edema, of which we had none.

## 7. Conclusions

This study confirms previous findings showing that CMR can offer a more thorough examination of patients with high PVC burden and a normal echocardiogram. This appears to be valid regardless of PVC morphology. CMR should therefore be considered in the diagnostic work–up. The prognostic importance of these findings needs to be assessed with longitudinal studies.

## Figures and Tables

**Figure 1 diagnostics-11-01505-f001:**
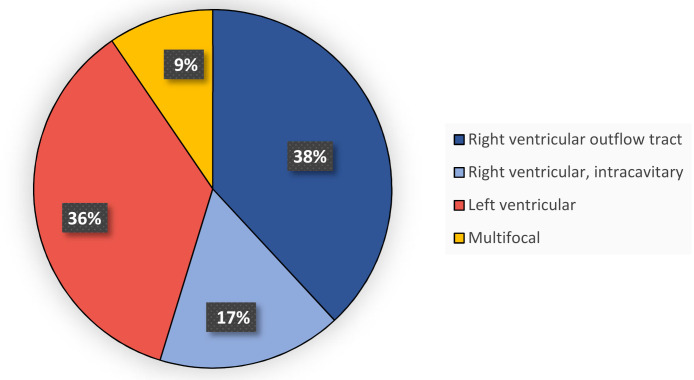
PVC morphology in 42 patients with PVC recorded on 12-lead ECG.

**Figure 2 diagnostics-11-01505-f002:**
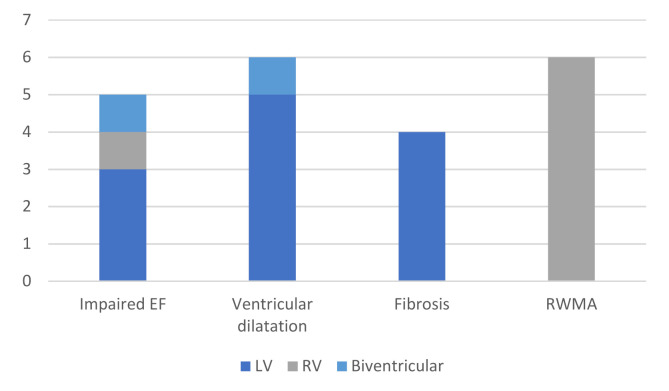
Summary of the CMR findings. LV = left ventricle, RV = right ventricle. RWMA = regional wall motion abnormalities.

**Figure 3 diagnostics-11-01505-f003:**
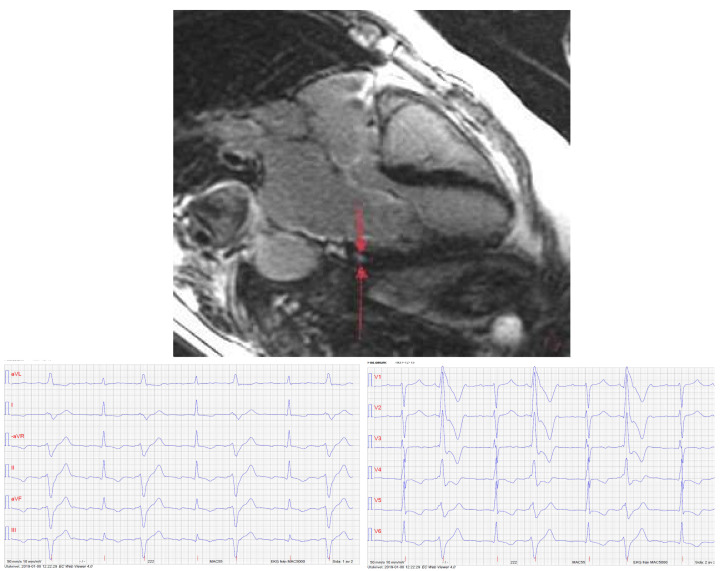
CMR image showing fibrosis in the basal inferolateral area of the left ventricle. Below: ECG from the same patient, showing frequent PVCs with a morphology corresponding to the fibrotic area, which is likely to act as site of origin.

**Table 1 diagnostics-11-01505-t001:** Baseline characteristics of 51 included patients.

	N	Percent (%)
Women	23	45
Hypertension	16	31
Paroxysmal Atrial Fibrillation	6	12
Diabetes Mellitus	1	2
Hyperlipidemia	3	6
<40 Years Old	10	20
40–70 Years Old	21	41
>70 Years Old	20	39

**Table 2 diagnostics-11-01505-t002:** Summary of the CMR parameters.

Parameter	Value
LV EF, %, median (IQR)	55 (53–58.5)
LVEDD, mL, mean (±SD)	165.43 (±33.72)
LVESD, mL, mean (±SD)	73.31 (±19.11)
LVSV, mL, median (IQR)	84.5 (76–107.5)
RVEF, mean (±SD)	56.72 (±5.65)
RVEDD, mL, mean (±SD)	141.95 (±39.55)
RVESD, mL, mean (±SD)	62.78 (±22.72)
RVSV, mL, median (IQR)	76.5 (73.5–106)

EF = ejection fraction, EDD = end diastolic diameter, ESD = end systolic diameter, SV = stroke volume.

**Table 3 diagnostics-11-01505-t003:** Site of origin of PVCs related to CMR findings.

CMR Finding	RVOT	RV, Intracavity	LV	Multifocal
Impaired Left Ventricular Function	1	0	0	1
Impaired Right Ventricular Function	1	0	0	0
Impaired Biventricular Function	1	0	0	0
Left Ventricular Dilatation	1	1	1	0
Biventricular Dilatation	0	0	1	0
Fibrosis	0	0	2	0
Wall Motion Abnormalities	3	1	1	0

Results from patients with CRM findings and PVCs recorded on 12-lead ECG.

**Table 4 diagnostics-11-01505-t004:** Prevalence of CMR findings in relation to site of origin of PVCs.

Origin of PVCs	Patients with Pathology at CMR	Patients with Normal CMR
RVOT	5	11
RV, Intracavity	2	5
LV	4	11
Multifocal	1	3

Data from 42 patients with PVCs recorded on 12-lead ECG.

**Table 5 diagnostics-11-01505-t005:** Site of origin of PVCs in relation to findings at CMR.

CMR Finding	Total Number	RV–PVCs	LV–PVCs	Multifocal PVCs	Unknown PVC Morphology
Any Solitary RV Finding	5	4	0	0	1
Any Solitary LV Finding	8	2	2	1	3
Any Biventricular Finding	3	1	2	0	0

RV = right ventricle, LV = left ventricle. Results from patients with CRM findings and PVCs recorded on 12-lead ECG.

## Data Availability

Data obtainable through the authors.

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
