# Peer review of "Magnetic Resonance Detects Structural Heart Disease in Patients with Frequent Ventricular Ectopy and Normal Echocardiographic Findings"

_diagnostics, 2021, doi:10.3390/diagnostics11081505_

Round 1

Reviewer 1 Report

The authors presented the paper "Magnetic Resonance detects structural heart disease in patients with frequent ventricular ectopy and normal echocardiographic findings"

Comments

1) The results in discussion section have to be compared with previously obtained results by other authors.

2) The conclusion section have to be rewritten to present the novel information from the obtained data from the authors. Moreover, the same recommendation is appropriate for the discussion section, too.

Minor comment

1) 2-5 year refences have to used in the Introduction section to present the recent material in the area.

2) The methods section may be have to be devided to subsections.

3) May be It is possible to use not so many different abbreviation in the text and in Tables but it is on your consideration.

Author Response

Dear Reviewer, 

thank you for your valuable comments. We have rewritten the Discussions and Conclusions sections according to your comment. More recent material in the field has been cited and included. Please see the new version of the manuscript.

We hope that you can find the changes relevant and our paper worth of publication.

Reviewer 2 Report

Congratulations on this study, which is really interesting and clinically important. The topic taken up by the authors is very important in the clinical context.

The study group is large enough to draw significant conclusions.

The research methodology is correct and presented accurately in details.

The results of the work are presented in a clear and understandable way.

The statistical methods used by the authors are appropriate.

The work is also very well prepared in terms of editing.

In my opinion, however, the results of the MRI should not be compared with classical echocardiographic parameters. Moreover, the choice of echocardiographic parameters to be assessed was quite limited. The authors wrote that they assessed ventricular  dimension, but did not write down exactly what echocardiographic parameters they used: LVDd measured by PLAX or left ventricular volume and stroke volume. If they have assessed the ventricular volumes and stroke volumes using MRI, the same parameters should be assessed by echocardiography. I find it very puzzling that the size of the left ventricle could be normal on echocardiography and enlarged on MRI. It is known that MRI is more accurate in assessing the right ventricle. However, for the assessment of the left ventricle, echocardiographic evaluation is already very accurate. Finally, if MRI assesses regional LV wall motion abnormalities, the same parameter should also be assessed in echocardiography. It would be worth assessing segmental left ventricular dysfunction by speckle tracking echocardiography.

Therefore, in my opinion, the authors should modify both the title and the content of the work. The study does not show "normal echocardiographic findings" but only  "normal basic echocardiographic findings".

More emphasis should be placed on the fact that left and right echocardiographic evaluation was limited and very basic.

After corrected these minor remarks above, in my opinion, the work is worth publication.

Author Response

Dear Reviewer, 

thank you for your valuable comments to our work. We agree with you that the results of the MRI should not be compared with classical echocardiographic parameters and that the choice of echocardiographic parameters to be assessed could be defined as limited. However, we wanted a strictly clinical cut for this study, and to carry it out in a setting where as an arrhytmologist (or cardiologist) you are to clinically evaualate a patient with a high PVC-burden and normal basic investigation (in this sense you are right in asserting that "basic echocardiographic findings" is probably more accurate) and start with the data you have access to. That is why we used the findings at echo as they were available at referral. 

In Sweden we have a dedicated medical speciality (Clinical Physiology) for medical diagnostics. All the echocardiograms in our study were performed and evaluated by a physician who is specialist in Clinical Physiology. This uses to mark the limit between echocardiography and basic echocardiography, which is more often performed by a general cardiologist in a clinical setting. 

Of course one could argue that more advanced echocardiographic parameters could have been used, but we wanted the study to reflect our clinical praxis in order to be clinically relevant and generalazible (at least in a swedish context). That is also why, for exemple, we chose to include consecutive patients regardless of PVC-morphology.

We thank you again for your comments. We hope we have made an argument for our choices and that you can find our paper worth of publication. 

Round 2

Reviewer 1 Report

Thank you for the revised version of the paper. However, I think that the conclusion section can be moderated to show better the novelty and to show the possible further investigation area.

Maybe, some more 2-5 year references have to be used in the Introduction section to present the recent material in the area.